# Providing person-centered care for patients with complex healthcare needs: A qualitative study

**Vincent J. T. Peters[1], Bert R. Meijboom****[1,2,3]\*, Jan Erik H. Bunt[4], Levinus A. Bok[5], Marianne W. van Steenbergen[6], J. Peter de Winter[7], Esther de Vries[2,8]**

1 Department of Management, Tilburg School of Economics and Management, Tilburg University, Tilburg, Noord-Brabant, The Netherlands, 2 Department of Tranzo, Tilburg School of Social and Behavioral Sciences, Tilburg University, Tilburg, Noord-Brabant, The Netherlands, 3 Department of Marketing, Innovation and Organization, Ghent University, Ghent, Belgium, 4 Department of Pediatrics, Elisabeth-Tweesteden Ziekenhuis, Tilburg, Noord-Brabant, The Netherlands, 5 Department of Pediatrics, Máxima Medisch Centrum, Veldhoven, Noord-Brabant, The Netherlands, 6 Department of Pediatrics, Jeroen Bosch Ziekenhuis, 's-Hertogenbosch, Noord-Brabant, The Netherlands, 7 Department of Pediatrics, Spaarne Gasthuis, Haarlem, Noord-Holland, The Netherlands, 8 Department of Jeroen Bosch Academy Research, Jeroen Bosch Ziekenhuis, 's-Hertogenbosch, Noord-Brabant, The Netherlands

\* b.r.meijboom@tilburguniversity.edu

## Abstract

### Background

People with chronic conditions have complex healthcare needs that lead to challenges for adequate healthcare provision. Current healthcare services do not always respond adequately to their needs. A modular perspective, in particular providing visualization of the modular service architecture, is promising for improving the responsiveness of healthcare services to the complex healthcare needs of people with chronic conditions. The modular service architecture provides a comprehensive representation of the components and modules of healthcare provision. In this study, we explore this further in a qualitative multiple case study on healthcare provision for children with Down syndrome in the Netherlands.

### Methods

Data collection for four cases involved 53 semi-structured interviews with healthcare professionals and 21 semi-structured interviews with patients (the parents of children with Down syndrome as proxy). In addition, we gathered data by means of practice observations and analysis of relevant documents. The interviews were audio-recorded, transcribed verbatim and analyzed utilizing the Miles and Huberman approach.

### Results

Our study shows that the perspectives on healthcare provision of professionals and patients differ substantially. The visualization of the modular service architecture that was based on the healthcare professionals' perspective provided a complete representation of (para)medical outcomes relevant to the professionals' own discipline. In contrast, the modular service

request from the Ethics Review Board of Tilburg University (contact via erb@tilburguniversity.edu) for researchers who meet the criteria for access to confidential data.

**Funding:** The author(s) received no specific funding for this work.

**Competing interests:** The authors have declared that no competing interests exist.

architecture based on the patients' perspective, which we define as a person-centered modular service architecture, provided a representation of the healthcare service that was primarily based on functional outcomes and the overall wellbeing of the patients.

## Conclusion

Our study shows that visualization of the modular service architecture can be a useful tool to better address the complex needs and requirements of people with a chronic condition. We suggest that a person-centered modular service architecture that focuses on functional outcomes and overall wellbeing, enables increased responsiveness of healthcare services to people with complex healthcare needs and provision of truly person-centered care.

## Background

An increasing number of people are living with complex healthcare needs resulting from multiple chronic conditions [1, 2]. This increase poses a challenge to adequate healthcare provision. Most healthcare services continue to focus predominantly on single diseases or prioritize medically oriented care (medical outcomes) over socially oriented care (functional outcomes). As a result, these healthcare services do not adequately respond to the complex healthcare needs of people with chronic conditions; current healthcare provision is not optimally tailored to their needs [1, 3]. Also from a societal perspective, it is important that healthcare services become more responsive to the complex needs and requirements of these people.

Down syndrome (DS), also known as trisomy 21, is the most common form of intellectual disability among newborn infants. At different ages, a variety of physical problems can arise and necessitate screening, prevention, and treatment [4–6]. The different health professions most frequently involved are pediatrics (celiac disease, growth, hypothyroidism, leukemia), cardiology (congenital heart defects), optometrist and ophthalmologist (visual acuity and squint), ENT-physician (chronic ear infections, hearing defect, and sleep apnea), orthopedics (hip dysplasia and dislocation), speech therapy (speech delay and disturbed oral motor function), dietetics (obesity and malnutrition), and physiotherapy (motor retardation and screening of development) [7, 8]. Although each separate clinical problem is well known, it is the personal tailoring of the screening, prevention and treatment in a patient with DS which makes the organization and delivery of person-centered care complex.

The complexity of healthcare services, an example of knowledge-intensive professional services, stems from multiplicity and diversity in their service offering [9–11]. Multiplicity refers to the growing number of involved providers, components and interactions in service provision [11, 12] and is demonstrated by the various professionals, from different units or departments, who deliver a high number of components for the treatment of patients with complex healthcare needs. This highly professionalized workforce needs to collaborate, something that could contradict the professional autonomy of the professionals [9, 13] and, consequently, increase the complexity of healthcare services. Diversity refers to the growing variety of providers, components and interactions that are required to fulfill diversified patient needs [11]. Each patient has an individual constellation and combination of health problems which implies that multiple professionals are required to address these health problems. In addition, the steep information asymmetry between professionals and patients, a characteristic that is inherent in knowledge-intensive professional services [9], can result in ambiguously expressed healthcare needs and increases the complexity of healthcare services.

An approach based on service modularity, a concept from the operations management domain, has the potential to reduce service complexity and increase responsiveness to complex healthcare needs [14]. Service modularity involves the decomposition of a complex service into modules and components. Modules are independent parts of a service with a specific function that can be offered individually, or in combination [15]. Within these modules, standardized components can be distinguished. These are the smallest elements in which a service can be meaningfully divided [16]. The decomposition of a complex service into modules and components is captured in the modular service architecture (MSA) and is defined as "the way that the functionalities of the service system are decomposed into individual functional elements to provide the overall services delivered by the system" [17 pp546]. The MSA is an intelligible visualization of all modules and components of a service and provides a comprehensive modular representation of a service offering [17]. It allows for the mixing-and-matching principle of modularity: (re)combining components and modules to create individualized modular packages. This principle ensures that each customer can be offered a selection of components and is treated as unique [3, 18]. As a result, services can be optimally tailored to the needs and preferences of individual customers.

Despite the potential of MSA to provide services that are responsive to the complex needs and requirements of customers, empirical evidence on the application of MSA is rare [19]. Although previous research provides examples of modular decomposition of healthcare services such as home care for the elderly [18], residential mental healthcare [20] and cancer care [21], these studies do not provide the complete MSA of these healthcare services. This results in an incomplete representation of the service offering and limits the potential of MSA to mix-and-match components and create truly individualized modular packages for each patient. Since only a few studies have addressed the applicability of MSA in complex services [13, 22, 23], there is still ambiguity around how to decompose a service offering into components and how to determine which of these components, alone or together, can be assigned as modules [24–26]. Dörbecker & Böhmann [27] have developed questions that can guide the identification of components and modules for the creation of MSA, but these are only applied to a limited extent [3]. In addition, the few studies that do address the applicability of MSA are traditionally conducted from the professional's perspective [13, 22, 23], which is surprising given the indispensable involvement of the customer in service provision [10, 12, 28].

In healthcare, the professional's perspective mainly reflects the provision of healthcare services aimed at improving medical outcomes [1, 13, 23] and does not respond to the individual situations of people with complex healthcare needs. As a result, care is often not optimally tailored to their needs. The medical outcomes are often not the most relevant from a patient's perspective; patients often attach greater value to functional outcomes and overall wellbeing [1, 2]. However, it is increasingly acknowledged that insight into the patient's perspective is becoming more important, especially for tailoring care to the needs and preferences of patients [1, 2, 29], quality of care [30, 31], and coordination of care [32, 33]. These are all considered essential elements of person-centered care [34]. Our aim is therefore twofold. First, we provide the complete modular service architecture of healthcare provision for people with complex healthcare needs. This allows for the creation of individualized modular healthcare packages and supports the provision of person-centered care. Second, we provide insight into the patient's perspective on MSA and explore how their perspective can support the provision of person-centered care. By doing so, we respond to the call for further empirical study on the application of MSA [19] and the call for more insight into the patients' perspective on complex (modular) healthcare services [19, 26, 34].

We address these gaps in a multiple case study where we explore the applicability of MSA in healthcare provision for people with complex healthcare needs. We explore this from the

perspective of patients as well as from that of the healthcare professionals. We used chronic healthcare for children with DS as an example, and focused on the question whether MSA can support the provision of person-centered care.

## Methods

### Ethical considerations

Ethical approval for this study was obtained from the Ethics Review Board of Tilburg University [EC-2017.60t]. Written informed consent was obtained prior to participation from all participants (the professionals and the parents of the children with DS).

### Study design

We carried out a qualitative multiple case study to explore the applicability of MSA in chronic healthcare provision. A multiple case study design was chosen because this enabled us to explore differences within as well as across cases [35]. The consolidated criteria for reporting qualitative research (COREQ) [36] were used as guideline for the study design and the data analysis (S1 File).

### Context

In the Netherlands, pediatric outpatient clinics organize multidisciplinary team appointments for children with DS, including a visit to medical, paramedical, and non-medical specialists, all on the same day [37]. These teams are called Downteams. We aimed to select a range of available Downteams in the Netherlands that vary in their composition and working methods in order to achieve a representative set of Downteams. We used purposive sampling logic and carefully selected four out of the 22 Downteams in the Netherlands [38] to include in our research. These four Downteams are well-known in the field and demonstrate variety in their composition, working methods and geographic location, resulting in a comprehensive view on chronic healthcare for children with DS. As such, they provided a good representation of all Downteams in the Netherlands.

### Participants

Recruitment of participants was carried out by the coordinators (the pediatricians) of the Downteams based on purposive sampling logic. In the summer of 2017, using e-mail, face-to-face requests, and telephone, they invited all the healthcare professionals in their Downteam and potentially interested parents of children with DS. The parents of the children with DS were considered as proxy for the children with DS (hereafter referred to as "patients"); this is common practice in pediatric research, especially in children with intellectual disability [39]. The e-mail included an invitation with a detailed explanation of the study. The potential participants were given as much time as needed to consider whether they wished to participate and, in the case of a positive decision, were asked to reply to the pediatrician and give consent for their contact details to be disclosed to the first author who then contacted the participants and scheduled the interviews. In total, 74 people agreed to participate; six people refused to participate due to time constraints (two professionals, four patients).

### Data collection

The data were collected by researcher VP through semi-structured interviews, observations and collecting documentation. From September 2017 until January 2018, 53 healthcare professionals and 21 patients were interviewed, each interview lasting from 45 to 75 minutes

**Table 1. Study participants.**

| Case A | Case B | Case C | Case D |
|---|---|---|---|
| ENT-doctor (2x) | ENT-doctor | Audiology assistant | Child psychologist |
| Dietician | Dietician | Contact parent | ENT-doctor |
| Doctor for the mentally handicapped (2x) | Doctor for the mentally handicapped | ENT-doctor | Doctor for the mentally handicapped |
| | | Dietician | |
| Ophthalmologist | Medical social worker | Doctor for the mentally handicapped | Occupational therapist |
| Pediatrician (2x) | Ophthalmologist | | Ophthalmologist |
| Parent (6x) | Orthoptist | Orthoptist | Pediatrician |
| Physiotherapist (2x) | Pediatrician (2x) | Ophthalmologist | Parent (5x) |
| Secretary | Parent (5x) | Pediatrician | Physiotherapist |
| Social worker | Physiotherapist | Parent (4x)c | Preverbal speech therapist |
| Speech therapist (2x) | Secretary | Physiotherapist | Secretary |
| | Specialized nurse | Secretary | Speech therapist |
| | Speech therapist | Speech therapist | |
| | Rehabilitation doctor | Social worker | |
| | Youth healthcare physician | | |

(Table 1). No significant changes in the health system or in staffing during the data collection period occurred. The interview questions were made up of a range of open-ended questions which aimed at an understanding of which healthcare elements were provided by each respective healthcare professional and helped us to acquire information on the patient's perspective on healthcare provision (S2 File). The same topics were discussed with both healthcare professionals and patients; questions were adapted to the perspective of the participant. Interviews were audio recorded and transcribed verbatim in a Word document. Data saturation was met after 65 interviews; however, for the sake of completeness the researchers agreed to perform the remaining scheduled interviews. Participants were asked to review their own transcript to improve the reliability of our interpretations; they provided additional information through follow-up emails.

Researcher VP also conducted 12 unstructured practice observations, three at each Downteam. Each observation lasted half a day and took place during a consultation of children with DS at a Downteam. Researcher VP followed a child with DS and their parents at each of their (consecutive) consultations (e.g. consultation with pediatrician, consultation with physiotherapist). This allowed us to get a better understanding of the daily practice of care provision. The observations focused on the questions "What elements of healthcare does the healthcare professional provide during the consultation and are there opportunities for patient input during healthcare provision?" Researcher VP made field notes and theoretical memos which helped to understand potential interpretations of the observations.

Last, researcher VP collected relevant documentation that was available both externally (e.g. national guideline [40], folders containing information about the Downteam) and internally (e.g. planning schemes, minutes from multidisciplinary meetings).

## Data analysis

The final data consisted of transcripts of the interviews, field notes and theoretical memos from the observations and documentation. The different types of data were complementary to each other: interviews helped us to acquire information on the professional's and patient's perspective on care provision, observations allowed us to get a better impression of the daily

practice of care provision, and the documents collected gave valuable information with regard to the composition and working methods of the Downteams. The data analysis was conducted in two stages: within-case and cross-case [41]. A thematic analysis of the content was carried out, using the three steps method described by Miles and Huberman [42]: 1) data reduction; 2) data display, and 3) drawing conclusions. The participants did not express themselves in modularity terms, but instead we used modularity as a perspective that guided interpretation of the data. By combining the information from the interviews, observations and documentation, we were able to describe and interpret the practices provided by healthcare professionals in modular terms. For example: we used the national guideline [40] to assign distinct parts of the consultation from each individual professional as modules, as per our definition of modules [15]. The transcripts, field notes and theoretical memos were then used to corroborate the parts assigned as modules. We used guiding questions (e.g. for what are the modules used?, who will use these modules?) [27] to validate our interpretation of modules. If this differed from our interpretation, we reconsidered how the modules had been assigned. As a result, we went back and forth with all the collected data. We returned to our participants to prevent potential errors of interpretation [43]. The participants recognized the modular perspective in their way of working. Analysis began with the coding of three interviews by one researcher (VP) using the initial coding scheme that was developed based on theoretical constructs. The codes were discussed among three researchers (VP, BM and EV). For the next ten interviews, two researchers (VP and BM) coded the interviews independently and then compared and discussed their codes. During this process, initial codes were altered and new codes were added. The three researchers (VP, BM and EV) discussed and assessed the outcomes of the coding until consensus was reached. The remaining interviews were then coded by one researcher (VP) using the final version of the coding scheme. The quotes from interviewees resulting from the analysis are presented in the text of the Results section; we illustrate the modular perspective in the quotes in square brackets.

## Results

### Within-case analysis

We created detailed descriptions for each of the four cases. Based on the information from the interviews, observations, and documentation, we described in modular terms the practices undertaken by the healthcare professionals in the four Downteams, using our coding of the text fragments as a basis. We assigned the distinct parts of the consultations from the various individual professionals as modules (e.g. Dietetic examination, Language production). The professionals explained that each module has specific meaning for their consultation and is based on the national DS guideline developed by the Dutch Pediatric Association [40].

> "*The healthcare parts* [modules] *I offer have specific meaning for the child and his/her parents*: *disorders, wellbeing and development.*" (Pediatrician A)

> "*My consultation is based on the national DS guideline and my discipline specific protocol. Those are the parts* [modules] *I offer during my timeslot.*" (Physiotherapist B)

We identified the components of the care currently provided by the professionals as elements of healthcare provision belonging to a certain module (e.g. Oral motor examination as an element of the module Dietetic examination, Analysis of used gestures as an element of the module Language production). These components are based on guidelines, protocols, and screening forms used in healthcare provision. For example, the ENT-doctor always evaluates

the throat (a module on its own) but does not provide all of the components potentially belonging to that module.

"*I do the mandatory screening for these children* [with DS]. *I do not have many options for mixing care parts* [modules] *related to my consultation, but sometimes I can leave out a small element* [component] *of the consultation.*" (ENT-doctor A)

"*Based on my screening form, I know which elements* [components] *belong to a specific care part* [module] *of my consultation. For example, if a patient suffers from celiac disease* [module], *certain elements* [components] *belong to that specific care part* [module]." (Dietician C)

Having allocated the different aspects of the professionals' work, it turned out that different individual modules contained identical components. Since the content of their healthcare provision was not prescribed in detail for each professional, different professionals ended up doing the same thing. For example, overlap occurred when two professionals both measured the height and weight of a patient and the professionals were not aware of this duplication. Also, at times professionals were under the impression that their colleagues were dealing with issues related to food and drink, for example. When these professionals met after their respective consultations, it turned out that none had dealt with those issues, with a resultant gap in healthcare provision. The MSA approach can assist in identifying overlaps and gaps in healthcare provision. Both professionals and patients expressed the need for this:

"*It would be great to remove duplicate elements* [components] *of our consultations. But I am not fully aware of what the other healthcare professionals do. In order to remove something* [components] *from my consultation, I need that insight. Otherwise those elements* [components] *might be missing.*" (Physiotherapist D)

"*We are used to discipline-oriented working, I hardly know what my colleagues are doing.*" (Physiotherapist A)

"*I am not aware of what, for example, a speech therapist can offer me during a consultation. And I am not the only parent facing this problem.*" (Parent B)

"[. . .] *It would be great to have some kind of overview of what we can expect from the Downteam* [. . .]" (Parent C)

We constructed the MSA visualizations based on the identified modules and components of each case. To illustrate this, the MSA of case A is shown in Fig 1. The MSAs of the other three cases are presented in S1 Fig.

Although the healthcare professionals did not express themselves in modular terms, they could recognize their way of working when presented with the MSA visualization. The MSAs based on the perspective of the healthcare professionals working in each Downteam show that the professionals are mainly focused on (para)medical conditions relevant to their own discipline. This led to consultations that are focused on (para)medical outcomes. This is not always the most relevant approach from the patients' perspective, as explained below.

## Cross-case analysis

For the cross-case analysis, we combined the detailed descriptions from each of the four cases. In each case, patients argued that current healthcare provision did not fully reflect their needs

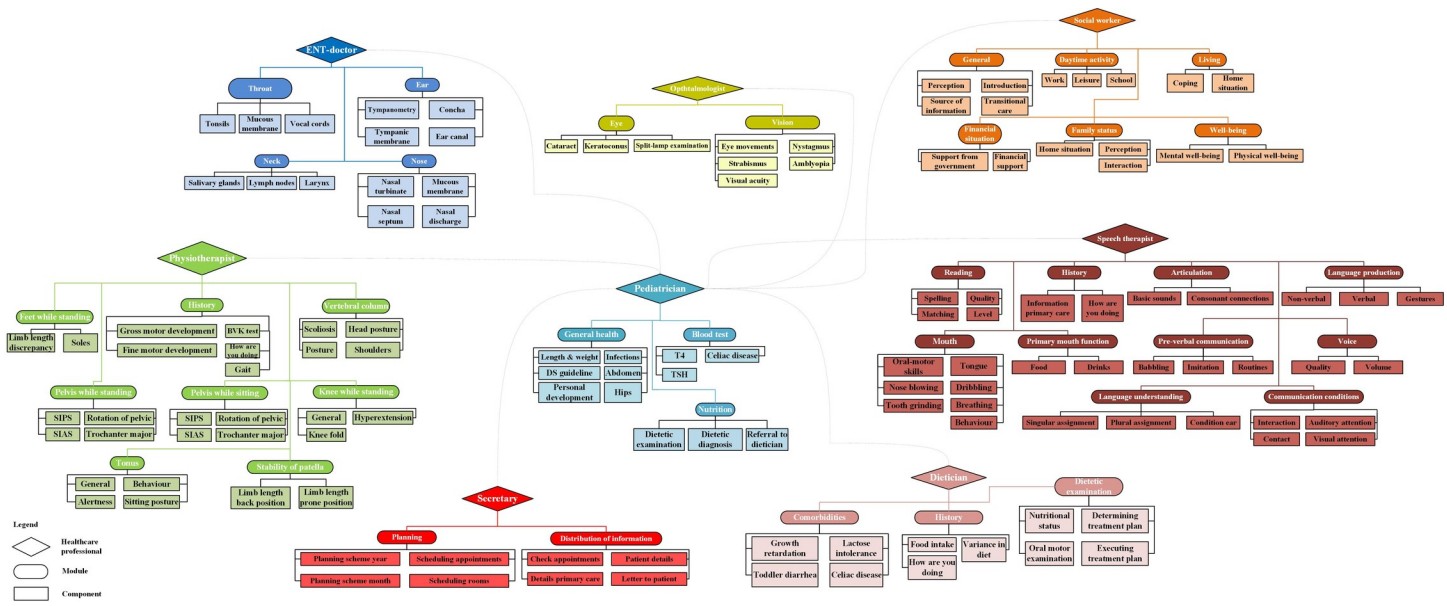

**Fig 1. Modular service architecture based on the healthcare professionals' perspective: Case A.**

and requirements. The patients actually attached greater value to functional outcomes and overall wellbeing as opposed to (para)medical outcomes.

> "[. . .] *I don't care which medication my child needs, I want him to get better and perform to the maximum of his capacity* [. . .]" (Parent C)

> "*The consultations are often not in line with what I require for my child. I do not know where to ask questions about eating and drinking.*" (Parent A)

> "*Sometimes I leave the Downteam and I still do not have the answers to my questions, as I did not know where to ask them.*" (Parent A)

> "*I do not understand why I always need to visit all the disciplines in the team. If my child has no problems related to the physiotherapist, why should we visit him? If I am not sick, I don't go to a general practitioner!*" (Parent B)

Therefore, we returned to our initial participants with the idea of presenting healthcare provision from the patients' perspective. We used the MSAs that were built from the healthcare professionals' perspective (Fig 1 and S1 Fig) as our starting point. The modules and components were reshaped in a way that reflected the intended patient needs i.e. functional outcomes and overall wellbeing. For instance, we suggested 'Participating in society' instead of 'Activities of daily living' and 'Getting rid of complaints' instead of 'Medical examination'. Interestingly, we observed a clear difference between the medical specialists and the other healthcare professionals. Paramedical specialists, non-medical specialists and patients were very enthusiastic about this approach. In particular, patients stressed that the reshaping might look like a minor difference, but that this was crucial for engaging in meaningful conversations with the healthcare professionals. It reflected the patients' actual needs and requirements.

"*Framing healthcare* [modules and components] *in a patient-centered way is mainly a different way of thinking, and does not necessarily change my way of working. If this is what patients want, I believe this is what we should offer.*" (Speech therapist C)

"[. . .] *this way of reframing healthcare for our child is fully recognizable and appealing* [. . .]" (Parent A)

"*This* [person-centered approach] *feels like we* [parents] *are being heard. Finally, we are not talking about what type of therapies my child needs, but what he is capable of.*" (Parent C)

The medical specialists were more reluctant. They expressed their concern about parents' capacity to know what is important to screen, because many problems are not easy to recognize based only on their symptomatology in DS.

"*An important function of our consultation is early detection of less desirable health situations that can occur more often in children with DS, without direct complaints (screening for problems to come). The question remains whether you can tackle these types of problems with demand-driven healthcare.*" (ENT-doctor D)

We dealt with this by engaging in conversations with the medical specialists and explaining to them that our suggestion does not imply changing their way of working, but rather changing their way of thinking: providing optimal healthcare to patients remains their responsibility. Presenting healthcare in a way that reflects patient needs and requirements does not harm the professional autonomy of medical specialists. It is a matter of changing presentation, not practice. These conversations helped to overcome the reluctance of the medical specialists.

The patients and professionals also reflected on the level of detail in which healthcare provision should be described. Patients argued that extensive descriptions of possible healthcare provision might cause them to lose track in the jungle of all possible components.

"[. . .] *I want to know what options I have before and during a consultation, but I do not need an extensive list. I need a sense of what I can expect or what I can ask* [. . .]" (Parent C)

"*With all due respect, I don't care what exact medical issue my child has. If I observe that his/her* [child] *skin is itchy, I want them* [healthcare professionals] *to get rid of the itch.*" (Parent A)

These comments inspired us to group individual components under umbrella headings. Components were only grouped if they fulfilled the same type of patient need. For example, we grouped components like 'Eczema' and 'Acne' under the umbrella component 'Skin disorder'. We also did this for the module 'Dealing with laws and regulations'. Specific types and forms of arrangements and regulations were grouped under four components 'Financial arrangements', 'Legal arrangements', 'Organizational arrangements', and 'Guardianship, administration and mentorship'. This made the MSA visualization more comprehensible for both patients and professionals.

Finally, we constructed the MSA visualization based on the patients' perspective (Fig 2). It takes the individual needs and requirements of children with DS as starting point for the provision of healthcare and not the fields of expertise of the healthcare professionals. In doing so, it focusses on the functional outcomes and overall wellbeing as opposed to the (para)medical outcomes.

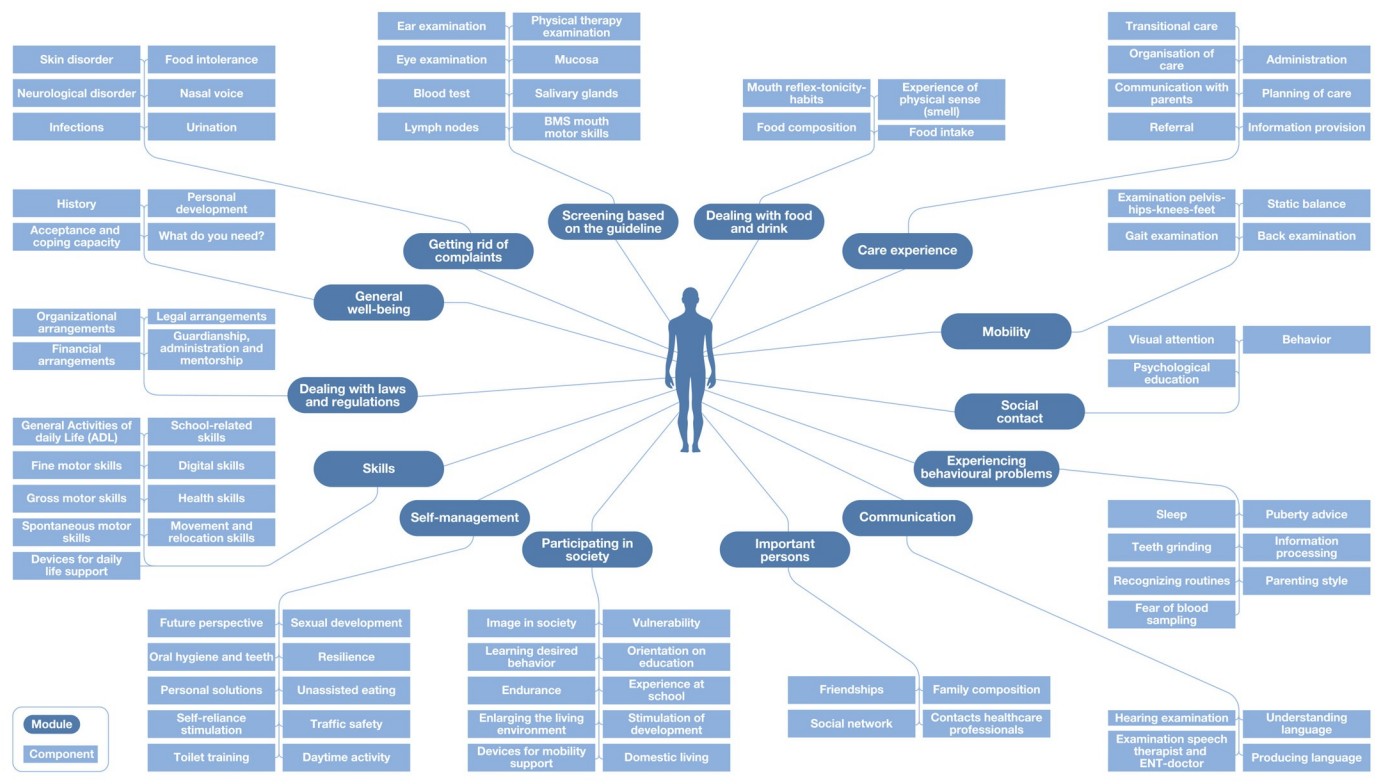

**Fig 2. Modular service architecture based on the patients' perspective.**

## Discussion

We explored the applicability of MSA visualization in chronic healthcare provision for children with DS from the perspective of patients alongside that of healthcare professionals. The modular perspective enabled us to fully decompose the healthcare provision into modules and components. Previous studies only provided partial modular decompositions of healthcare services [18, 20, 21] which limited the potential of modularity to reduce service complexity and increase responsiveness to complex healthcare needs [14]. Our results show that the MSA proved to be very illuminating for professionals and patients since it led to insight into the work practices of each professional, which increased transparency on services offered for both professionals and patients. In addition, the MSA revealed gaps and overlaps in healthcare provision, and provided opportunities to deal with unnecessary duplications and blind spots. Moreover, we show that MSA visualization provides possibilities for mixing and matching components and modules to address individual needs and, as such, increases the responsiveness of healthcare services to people with complex healthcare needs. This demonstrates that MSA supports service customization [10, 16, 18], which can consequently lead to truly person-centered care provision [26, 34]. As such, the MSA visualization provides a means of dealing with the complexity (i.e. multiplicity and diversity) of knowledge intensive professional services [9, 11]. As distinct from previous studies on MSA [13, 22, 23], we have incorporated the patients' perspective on MSA into our study and show that their perspective is essential for fulfilling the needs and preferences that are considered relevant by patients. The indispensable role of customers has been acknowledged in the extant service literature [e.g. 10, 12, 28], but

the literature on service modularity has mostly overlooked this [19]. Our study shows that the patient's perspective is essential to mix-and-match components in such a way that modular healthcare packages are created that are truly responsive to the needs and requirements of people with complex healthcare needs. As such, the service offering can be customized effectively [19] and the provision of person-centered care is supported [26, 34]. Furthermore, the results revealed the similar and contrasting viewpoints of healthcare professionals and patients.

We show that the MSA built from the perspective of the patients differs substantially from the MSA built from the perspective of the healthcare professionals. The MSA based on the perspective of the professionals provided a complete representation of the healthcare service based on (para)medical outcomes relevant to their own discipline: they focus on 'What-can-we-offer?' As such, it is an example of more traditionally oriented healthcare organized around single diseases within separate silos [44]. This introduces the risk that healthcare provision is focused on the (para)medical outcomes of the separate diseases instead of functional outcomes for the patients. This is in accordance with the findings of other researchers [1, 2, 6, 31, 45]. Current developments, however, focus more and more on the needs and requirements that are considered relevant by patients. In other words, the needs and requirements of patients with complex healthcare needs should serve as the starting point for their healthcare provision [1, 5].

The MSA that is based on the perspective of patients represents healthcare provision in a more person-centered way, as it focuses on 'What-do-I-need?'. For example, we reorganized components as 'Enlarging the living environment', 'Experience at school' as part of the person-centered module 'Participating in society' and the components 'Traffic safety' and 'Sexual development' as part of the person-centered module 'Self-management'. This person-centered MSA visualization provides a complete representation of the healthcare service based on functional outcomes and overall wellbeing and shows that insight into the patients' perspective is important for the delivery of person-centered care [30, 31, 33]. While previous studies on modular decompositions implicitly assume that they fulfill patients' needs and preferences [18, 20], we show that the person-centered MSA can be used as a tool to ensure the complex healthcare needs of people with chronic conditions are fulfilled. It offers patients and professionals the possibility of mixing and matching person-centered modules and components to create individualized person-centered care packages without ignoring the professional role of the healthcare professionals. The extent to which each patient can create their own modular package is debatable: while some patients are clearly capable of this, it may be more difficult for others [9, 13]. It would be difficult for patients with limited advocacy skills to create and arrange their own healthcare services [31]. The person-centered MSA ensures that it is the patients' needs that guide medical decisions and implies that each patient can be offered a modular package that fits with their needs and requirements [23], a promising development for people with complex healthcare needs, that is not yet standard practice [6].

For healthcare professionals, it can be challenging to deliver person-centered care. MSA can serve as a tool to increase their understanding of people's complex healthcare needs and identify duplications and gaps in their healthcare provision. The MSA also helps to remind them why they are in the caring profession and how they can provide patients with what they want and need. Previous research has shown that tools, like care mapping, have the potential to support the provision of person-centered care [46–48]. The MSA encourages reflection on the working methods of healthcare professionals and draws attention to the social situation of a patient, enabling healthcare professionals to provide person-centered care. Our person-centered MSA approach can be applied by others by following three steps: 1) detailed identification of all individual healthcare parts and elements (modules and components) in collaboration with patients and professionals, 2) labelling and reshaping these parts from the

patients' perspective, thereby focusing on functional outcomes and overall wellbeing when combining and grouping components and modules, and 3) selection of appropriate modules and components for person-centered healthcare provision. Previous research has shown that–once established–applying a person-centered approach does not require additional time from professionals; it even leads to more efficient care [49]. Our proposed person-centered approach provides more clarity on how to identify the individual parts of a service offering and which part(s) can be considered as components or as modules, which is crucial for the modular decomposition of services [24–27]. By applying our approach, future studies can demonstrate the modular composition of their case under study and the insights obtained can become more relevant for theory as well as for practice.

We have not yet implemented our findings in one or more of the Downteams under study. Doing this would be the next step, as would conducting a follow-up study to show whether the person-centered approach is truly feasible in current healthcare settings. Our suggested person-centered MSA approach can also be used as a basis for future healthcare design [50]. These findings could be applicable to other patient groups with complex healthcare needs (e.g. diabetes, oncology, geriatrics) with little adjustments.

Our study has some limitations. First, the results were obtained in the Downteams of Dutch hospitals. Interpretations for other patient groups with complex healthcare needs is dependent on the similarity between their needs on an organizational level. We believe that the MSA approach is also applicable for patients with more variable multi-morbidity, but a similarity in their healthcare needs, such as patients with cancer. Cancer is a complex condition manifesting in many different forms for which treatment usually requires various combinations of chemotherapy, radiotherapy, immunotherapy and/or surgery, leading to different forms and differing severity of side effects of the treatment options [21, 51]. However, on an organizational level, there is much similarity in cancer treatment. Therefore, MSA approaches are also likely to be useful for cancer patients. MSA can be useful in such situations because it ensures that professionals are aware of the full range of care and service components and patients are fully informed about treatment and support options. We also believe that the MSA approach is applicable in many other types of complex services such as legal services or higher educational services. For example, when clients face a legal conflict (divorce, termination of employment etc.) they can make use of a variety of providers in dealing with their conflict. Each provider is responsible for providing a subset of services for the client and collectively the providers offer the service that fits with the client's needs and wishes. Approaching legal services from a modular perspective allows for the decomposition into components and modules, resulting in transparency on the supply side of legal services. The modular perspective is also relevant for clients because although accurate information about legal services becomes increasingly available online [52], the clients are not fully aware of what each legal provider can offer them. This results in legal services that are not completely tuned to the needs and requirements of clients. In higher educational services in many countries there has been an increasing focus on individualized instructions, despite the increasing number of students [53]. As a result, there is a need to make higher education services available to large number of students and, at the same time, offer an individualized learning package for each student. A modular approach could help in dealing with this issue, as it provides opportunities to offer curricula or interdisciplinary programs that are designed based on modules, where each student's program is tailored to their individual needs and wishes [54]. Future research should test whether the MSA approach is feasible in these complex service settings. Second, we did not include healthcare professionals from primary care (e.g. general practitioner, youth health care physician) in our study, because they have a very limited role in chronic DS healthcare in the Netherlands. This could be different for other chronic diseases and countries. Third, parents were considered as proxy

for the children with DS in our study. Although parents are often used as proxy in pediatric care, differences between children and parent proxy have been described [39].

Future research could include the perspectives of healthcare professionals from primary care and children themselves in order to fully capture the modular perspective on chronic healthcare provision for people with complex healthcare needs. Furthermore, future studies are required to address the coordination of our person-centered MSA approach. A lack of coordination could lead to increased health risks for people with complex healthcare needs, for instance when patients receive conflicting treatments or unnecessary duplications from multiple healthcare professionals. Coordination of healthcare is, therefore, of great importance for people with complex healthcare needs [1, 32]. In modular healthcare services, coordination is achieved by interfaces. Interfaces allow for the interaction and communication between modules, components, and people (patients and professionals) involved in healthcare provision [55]. These interfaces can provide a tight fit between modules, components and people and, as such, can reduce the risk of conflicting treatments or unnecessary duplication in healthcare provision. Further research should address the role of interfaces in healthcare provision for people with complex healthcare needs. Lastly, we did not measure the value of the MSA approach. Future research could examine whether Downteams with a person-centered MSA approach are associated with better outcomes on process-indicators (e.g. adherence to guidelines, access to care) and outcome indicators (e.g. safety of care, patient satisfaction) compared with Downteams that use a traditional approach.

## Conclusion

We performed a qualitative multiple case study to explore the applicability of MSA visualization in healthcare provision for people with complex healthcare needs, using chronic healthcare for children with DS as our proof-of-concept. To our knowledge, this is the first empirical study that explores the applicability of MSA in healthcare services from the perspective of the patients besides that of the healthcare professionals. Our modular perspective allowed us to provide a complete representation of their healthcare provision. Our reshaping of the results into a person-centered MSA visualization, focusing on functional outcomes and overall well-being instead of (para)medical outcomes of separate disease entities, enables provision of truly person-centered care. This person-centered MSA approach can thereby contribute to increased responsiveness of healthcare services for people with complex healthcare needs.

## Supporting information

**S1 Fig. Modular service architecture based on the healthcare professionals' perspective: Case B-D.**
(PDF)

**S1 File. A 32-item checklist for reporting qualitative studies (COREQ).**
(DOCX)

**S2 File. Topic list.**
(DOCX)

## Acknowledgments

The authors would like to thank all the parents of children with Down syndrome and the healthcare professionals involved for their kind participation in the study.

## Author Contributions

**Conceptualization:** Vincent J. T. Peters, Bert R. Meijboom, Esther de Vries.

**Data curation:** Vincent J. T. Peters, Bert R. Meijboom, Esther de Vries.

**Formal analysis:** Vincent J. T. Peters, Esther de Vries.

**Investigation:** Vincent J. T. Peters.

**Methodology:** Vincent J. T. Peters, Bert R. Meijboom, Esther de Vries.

**Resources:** Jan Erik H. Bunt, Levinus A. Bok, Marianne W. van Steenbergen, J. Peter de Winter.

**Supervision:** Bert R. Meijboom, Esther de Vries.

**Validation:** Jan Erik H. Bunt, Levinus A. Bok, Marianne W. van Steenbergen, J. Peter de Winter.

**Visualization:** Vincent J. T. Peters.

**Writing – original draft:** Vincent J. T. Peters.

**Writing – review & editing:** Vincent J. T. Peters, Bert R. Meijboom, Jan Erik H. Bunt, Levinus A. Bok, Marianne W. van Steenbergen, J. Peter de Winter, Esther de Vries.

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
