## [Decision Letter · Decision Letter 0]

24 Jul 2020

PONE-D-20-03347

Providing person-centered care for patients with complex healthcare needs: A qualitative study

PLOS ONE

Dear Dr. Meijboom,

Thank you for submitting your manuscript to PLOS ONE. After careful consideration, we feel that it has merit but does not fully meet PLOS ONE’s publication criteria as it currently stands. Therefore, we invite you to submit a revised version of the manuscript that addresses the points raised during the review process.

We look forward to receiving your revised manuscript.

Kind regards,

Janhavi Ajit Vaingankar

Academic Editor

PLOS ONE

Journal Requirements:

2. Please provide the dates of participant recruitment in your manuscript.

4. Please amend your manuscript to include your abstract after the title page.

Reviewers' comments:

Reviewer's Responses to Questions

**Comments to the Author**

1. Is the manuscript technically sound, and do the data support the conclusions?

Reviewer #1: Yes

Reviewer #2: Partly

2. Has the statistical analysis been performed appropriately and rigorously? 

Reviewer #1: N/A

Reviewer #2: N/A

3. Have the authors made all data underlying the findings in their manuscript fully available?

Reviewer #1: Yes

Reviewer #2: Yes

4. Is the manuscript presented in an intelligible fashion and written in standard English?

Reviewer #1: Yes

Reviewer #2: Yes

5. Review Comments to the Author

Reviewer #1: Overall

Overall this is a well written paper reporting original and important results, highly suitable for publication with some revision as indicated. The study design is appropriate and well described. Adherence to and inclusion of the COREQ is a strength. More could be explored in the Discussion section about of the differences between health professional and parent respondents, and applicability or otherwise of results to other settings.

Background

The concept and the term ‘modular service architecture’ is unlikely to be familiar to most readers of the journal. Add in more detail here about the principles that underpin MSA and justify more fully why and how these principles are applicable to health care for those with complex conditions.

More detail is also required about DS as a complex condition in the introduction. Although a complex condition with heterogeneous manifestations, nevertheless the constellation and combination of functional problems are reasonably predictable for this group. See comment about following this up under Discussion below.

Methods

How and why were the four Downteams selected as cases? Are they typical/atypical of all such teams? What diversity do they display?

Over what time period were all the interviews undertaken? The first 65 (reaching data saturation) and then the remainder? Were there any significant changes in the health system or staffing during the data collection period?

Provide more detail about how the collection and analysis of the direct observational data was undertaken. How did this feed into the within-case analysis?

Results

The figures provided are excellent, if complex. High quality reproduction of these is going to be crucial to readability – this has been mentioned in comments to the editor.

It would be worth pointing out in a little more detail in the text where the most striking gaps and duplication in care have been exposed by the research.

With regard to quotes – excellent to see the different health professional (HP) respondents identified by discipline but please also assign a unique identifier to each respondent. e.g. dietitian A; physiotherapist B – so we as readers know it is not merely the same dietitian being quoted throughout. And same again, with each Parent respondent – Parent a, Parent b, and so on.

Were there clear differences between the medical respondents and other HP respondents? Was there variation within these sub-groups? Clarify in the text.

Discussion

Referring to the point about DS as a particular sort of complex condition, compare this briefly to the patients with widely variable multi-morbidity, where there is not so often a common origin for the constellation of problems, and functional issues will be far more widely diverse. MSA approaches are likely more difficult to apply in such settings, so expand your explanation in the limitations paragraphs to cover this more clearly. Perhaps such an approach is only feasible for a particular, well-defined complex condition? Or not?

One of your key findings is around the stark difference between the patient and health professional perspectives. This deserves more prominence in the Discussion, since your main audience are health professionals who may find this challenging. You could refer more widely to other studies in different contexts – e.g.

Young, J., Poole, U., Mohamed, F., Jian, S., Williamson, M., Ross, J., Jaye, C., Radue, P., & Egan, T. (2019). Exploring the value of social network 'care maps' in the provision of long-term conditions care. Chronic Illness. Advance online publication. doi: 10.1177/1742395319836463

Abstract

Generally clear and an accurate representation of the paper, but add in a few more words about what MSA is, in line with the request about this in the Background section above.

Style/writing/references

In the main, the paper is structurally sound and reads well. Check again throughout for word choice and grammatically correct English. I have indicated below some of the more obvious corrections needed:

Line 51 – ‘Service modularity involves the decomposition of a complex service into modules…’ – the word ‘decomposition’ is probably better replaced with ‘deconstruction’, but check meaning and English language for clarity

Line 91 – ‘These four Downteams are well known in the field and provide a good of all Downteams in the Netherlands,…’ - correct the sentence

Line 109 – ‘The data were collected by VP…’ – explain who VP is – e.g. ‘author VP’ or ‘researcher VP’

Line 141 – ‘Three researched discussed and assessed the…’ – correct the sentence

Line 272-3 – ‘The modular perspective enabled us to fully decompose the healthcare provision into modules…’ – correct the word ‘decompose’ (similar problem to Line 51 comment above)

Reviewer #2: The paper started to address an interesting and timely research area, focusing on providing person-centered care for patients with complex healthcare needs. The authors draw on some prior studies, but do not offer a very critical literature analysis. This is needed to strengthen the paper’s argument and draw out the gaps they seek to address. Also, the paper needs to present a much deeper methods and findings sections as well as stronger discussion and conclusion sections in order to offer value to the reader. Overall, the manuscript makes some very interesting points and I realise that a lot of work went into this study. Nevertheless, I see room for improvement which will help to enhance clarity, readability, practical and theoretical contributions. The following paragraphs address each section of the paper in more detail and provide suggestions on how to revise the paper.

Major concerns:

Introduction:

While the author(s) establish some links to some extant literature, author(s) need to establish a more coherent framework for the overall paper. That means, the introduction should clearly indicate the need for this paper in relation to extant research studies. Which gap(s) are authors trying to address here? In order to link back to relevant extant literature, work on services, service complexity (Kreye et al. and work by Andy Neely et al.) and customization/modularization needs to be further emphasized (which gaps you seek to address and what you found). This would help to further ground your study in extant work and help you to motivate your study further.

Conceptual background & Theoretical development

The authors need to establish some clearer links to extant service literature (e.g. please see suggested key references). The author(s) should clearly draw out what is known for their key concepts. For instance, who is responsible for breaking up modules and components in your setting?; How easily can this be done?; How would this translate to other setting and what does prior work say about all this? This section needs much more information and guidance for reader to position the gaps in prior work. Also, given the nature of the services you are investigating, the authors should at least acknowledge prior work on knowledge intensive professional (KIP) services and their unique characteristics and features.

Methods and Analysis:

Overall, this section started to offer some insights, but I see room for improvements to obtain robust results and strengthen the paper’s argument. The authors should address the following concerns:

# Please clearly tell the reader about your sampling logic with regards to interviewees, context/sector and type of services.

# here you need to also bring out the differences between interviewees/observations etc. At the moment, most of the data converge and no real differences are noticeable.

# Clearly and in more detail explain your data analysis steps (what have you analysed first and how)

# A lot of the results section “tells” reader about your data rather than “shows” your data. Here, some data tables/figures and much more of a narrative would help to convince the reader that your findings support your conclusions.

# What is the impact of different skills/capabilities of professionals when talking about/judging modules and components?

# Parents may struggle to judge (as we are talking about KIP services) the value of medical services. How can you really compare data from healthcare professionals and non-professionals? Wouldn’t you expect differences in perception and value?

# How have you measured the value (e.g. information exchange) of new/revised MSAs?

Discussions and Conclusions:

Derived from a conceptual background section which did not clearly draw out the gaps the paper seeks to address, the discussion and conclusion sections do offer only some additional value to the reader as it stands. The authors need to offer more fine-grained results here and discuss what they intended to find out in the introduction section (link to research question; overall aim of the paper). The authors should also link their findings to prior studies. Overall, the authors need to draw out clearly what the theoretical contributions are and how they add to the existing body of knowledge. This section also needs to clear link back to extant studies to offer some clear value to the reader. This should then help to derive some more insightful practice and policy implications.

Useful references:

Kreye et al. (2015). Servitizing manufacturers: The importance of service complexity and contractual and relational capabilities. Production Planning & Control, Vol. 26 No. 14-15, pp. 1233-1246.

Neely, A., D. McFarlane, and I. Visnjic 2011. “Complex Service Systems – Identifying Drivers, Characteristics and Success Factors.” 18th European Operations Management Association Conference, Cambridge, UK, July.

6. PLOS authors have the option to publish the peer review history of their article (what does this mean?). If published, this will include your full peer review and any attached files.

Reviewer #1: **Yes: **Professor Sue Pullon

Reviewer #2: No

---

## [Author Response · Author response to Decision Letter 0]

2 Sep 2020

Academic editor: Janhavi Ajit Vaingankar

Journal Requirements:

Q1: Please ensure that your manuscript meets PLOS ONE's style requirements, including those for file naming. The PLOS ONE style templates can be found at

R: We have ensured that our manuscript meets PLOS ONE’s style requirements, including those for file naming. Please see our changes throughout the manuscript.

Q2: Please provide the dates of participant recruitment in your manuscript.

R: We have added the dates of participant recruitment in the manuscript

C: Recruitment of participants was carried out by the coordinators (the pediatricians) of the Downteams based on purposive sampling logic. In the summer of 2017, they invited all the healthcare professionals in their Downteam and potentially interested parents of children with DS using e-mail, face-to-face requests and telephone. (Methodology, page 10)

Q3: We note that you have indicated that data from this study are available upon request. PLOS only allows data to be available upon request if there are legal or ethical restrictions on sharing data publicly. For information on unacceptable data access restrictions, please see http://journals.plos.org/plosone/s/data-availability#loc-unacceptable-data-access-restrictions.

R: We have added more detail on why there are ethical and legal restrictions on sharing our data publicly.

C: Data cannot be shared publicly because the data contain potentially identifying or sensitive patient information that cannot be de-identified. Data are available upon request from the Ethics Review Board of Tilburg University (contact via erb@tilburguniversity.edu) for researchers who meet the criteria for access to confidential data.

Q4: Please amend your manuscript to include your abstract after the title page.

R: We have amended the manuscript and included the abstract after the title page. Please see the manuscript accordingly.

Reviewer 1: Professor Sue Pullon

Q5: Overall this is a well written paper reporting original and important results, highly suitable for publication with some revision as indicated. The study design is appropriate and well described. Adherence to and inclusion of the COREQ is a strength. More could be explored in the Discussion section about of the differences between health professional and parent respondents, and applicability or otherwise of results to other settings.

R: We thank the reviewer for her compliment. We will elaborate on her points in more detail in the points below.

Background

Q6: The concept and the term ‘modular service architecture’ is unlikely to be familiar to most readers of the journal. Add in more detail here about the principles that underpin MSA and justify more fully why and how these principles are applicable to health care for those with complex conditions.

R: We added more detail about the principle that underpins MSA and justify more fully why and how these principles are applicable to healthcare for those with complex conditions.

C: The decomposition of a complex service into modules and components is captured in the modular service architecture (MSA) and is defined as “the way that the functionalities of the service system are decomposed into individual functional elements to provide the overall services delivered by the system” [17 pp546]. The MSA is an intelligible visualization of all modules and components of a service and provides a comprehensive modular representation of a service offering [17]. It allows for the mixing-and-matching principle of modularity: (re)combining components and modules to create individualized modular packages. This principle ensures that each customer can be offered a selection of components and is treated as unique [3,18]. As a result, services can be optimally tailored to the needs and preferences of individual customers. (Background, page 6)

Q7: More detail is also required about DS as a complex condition in the introduction. Although a complex condition with heterogeneous manifestations, nevertheless the constellation and combination of functional problems are reasonably predictable for this group. See comment about following this up under Discussion below.

R: We added more detail about DS as a complex condition.

C: Down syndrome (DS), also known as trisomy 21, is the most common form of intellectual disability among newborn infants. At different ages, a variety of physical problems can arise and necessitate screening, prevention, and treatment [4-6]. The different health professions most frequently involved are pediatrics (celiac disease, growth, hypothyroidism, leukemia), cardiology (congenital heart defects), optometrist and ophthalmologist (visual acuity and squint), ENT-physician (chronic ear infections, hearing defect, and sleep apnea), orthopedics (hip dysplasia and dislocation), speech therapist (speech delay and disturbed oral motor function), dietetics (obesity and malnutrition), and physiotherapy (motor retardation and screening of development) [7,8]. Although each separate clinical problem is well known, it is the personal tailoring of the screening, prevention and treatment in a patient with DS which makes the organization and delivery of person-centered care complex. (Background, page 4-5)

Additional reference:

5. Skotko BG, Davidson EJ, Weintraub GS. Contributions of a specialty clinic for children and adolescents with Down syndrome. American Journal of Medical Genetics Part A. 2013;161(3): 430–437. doi:10.1002/ajmg.a.35795.

Methods

Q8: How and why were the four Downteams selected as cases? Are they typical/atypical of all such teams? What diversity do they display?

R: We have added more detail about how and why the four Downteams were selected as cases in our research.

C: We aimed to select a range of available Downteams in the Netherlands that vary in their composition and working methods in order to select a representative set of Downteams. We used purposive sampling logic and carefully selected four out of the 22 Downteams in the Netherlands [38] to include in our research. These four Downteams are well-known in the field and demonstrate variety in their composition, working methods and geographic location, resulting in a comprehensive view on chronic healthcare for children with DS. As such, they provided a good representation of all Downteams in the Netherlands. (Methods, page 8-9)

Q9: Over what time period were all the interviews undertaken? The first 65 (reaching data saturation) and then the remainder? Were there any significant changes in the health system or staffing during the data collection period?

R: The interviews were undertaken from September 2017 until January 2018. All interviews were scheduled in advance and for the purpose of completeness, we decided to undertake all the interviews. No significant changes in the health system or in staffing during the data collection period occurred.

C: From September 2017 until January 2018, 53 healthcare professionals and 21 patients were interviewed, each interview lasting from 45 to 75 minutes (Table 1). No significant changes in the health system or in staffing during the data collection period occurred. (Methods, page 10)

Q10: Provide more detail about how the collection and analysis of the direct observational data was undertaken. How did this feed into the within-case analysis?

R: We have added more detail about how the collection and analysis of the direct observational data was undertaken.

C Researcher VP followed a child with DS and their parents at each of their (consecutive) consultations (e.g. consultation with pediatrician, consultation with physiotherapist). This allowed us to get a better understanding of the daily practice of care provision. (Methods, page 11)

Researcher VP made field notes and theoretical memos which helped to understand potential interpretations of the observations. (Methods, page 11)

Results

Q11: The figures provided are excellent, if complex. High quality reproduction of these is going to be crucial to readability – this has been mentioned in comments to the editor.

R: We thank the reviewer for the compliment and have used the Preflight Analysis and Conversion Engine (PACE) digital diagnostic tool to ensure that figures meet PLOS requirements.

Q12: It would be worth pointing out in a little more detail in the text where the most striking gaps and duplication in care have been exposed by the research.

R: We have added an additional example in the manuscript about where a striking gap in care has been exposed.

C: Also, at times professionals were under the impression that their colleagues were dealing with issues related to food and drink, for example. When these professionals met after their respective consultations, it turned out that none had dealt with those issues, with a resultant gap in healthcare provision. The MSA approach can assist in identifying overlaps and gaps in healthcare provision. (Results, page 14)

Q13: With regard to quotes – excellent to see the different health professional (HP) respondents identified by discipline but please also assign a unique identifier to each respondent. e.g. dietitian A; physiotherapist B – so we as readers know it is not merely the same dietitian being quoted throughout. And same again, with each Parent respondent – Parent a, Parent b, and so on.

R: We thank the reviewer for this suggestion and have assigned unique identifiers to each respondent throughout the manuscript. For clarity sake, we have only added a single example in this response letter.

C: “The healthcare parts [modules] I offer have specific meaning for the child and his/her parents: disorders, well-being and development.” (Pediatrician A) (Results, page 13)

Q14: Were there clear differences between the medical respondents and other HP respondents? Was there variation within these sub-groups? Clarify in the text.

R: We observed a clear difference between the medical respondents and the other healthcare professional respondents. We added a few lines emphasizing this.

C: Interestingly, we observed a clear difference between the medical specialists and the other healthcare professionals. Paramedical specialists, non-medical specialists and patients were very enthusiastic about this approach. (Results, page 16-17)

The medical specialists were more reluctant. They expressed their concern about parents’ capacity to know what is important to screen, because many problems are not easy to recognize based only on their symptomatology in DS. (Results, page 17)

Discussion

Q15: Referring to the point about DS as a particular sort of complex condition, compare this briefly to the patients with widely variable multi-morbidity, where there is not so often a common origin for the constellation of problems, and functional issues will be far more widely diverse. MSA approaches are likely more difficult to apply in such settings, so expand your explanation in the limitations paragraphs to cover this more clearly. Perhaps such an approach is only feasible for a particular, well-defined complex condition? Or not?

R: We expanded the explanation in the limitations paragraph. We belief that MSA approaches are feasible for patients with widely variable multi-morbidity when there is a similarity in the multiplicity and diversity in the patients’ needs, reflected in the various types of providers and organizations involved in their healthcare provision. We provide an example of patients with cancer. Please see the manuscript accordingly.

C: Interpretations for other patient groups with complex healthcare needs is dependent on the similarity between their needs on an organizational level. We believe that the MSA approach is also applicable for patients with more variable multi-morbidity, but a similarity in their healthcare needs, such as patients with cancer. Cancer is a complex condition manifesting in many different forms for which treatment usually requires various combinations of chemotherapy, radiotherapy, immunotherapy and/or surgery, leading to different forms and differing severity of side effects of the treatment options [21,51]. However, on an organizational level, there is much similarity in cancer treatment. Therefore, MSA approaches are also likely to be useful for cancer patients. MSA can be useful in such situations because it ensures that professionals are aware of the full range of care and service components and patients are fully informed about treatment and support options. (Discussion, page 22-23)

Additional references:

21. Gobbi C, Hsuan J. Modularity in cancer care provision. In: van Donk PD, de Koster R, de Leeuw S, Fransoo J, van der Veen J, editors. EurOMA 2012: Proceedings of the 19th European Operations Management Association (EurOMA); 2012 Jul 1-5; Amsterdam, the Netherlands. Brussels: European Management Association.

51. Cortis LJ, Ward PR, McKinnon RA, Koczwara B. Integrated care in cancer: What is it, how is it used and where are the gaps? A textual narrative literature synthesis. European Journal of Cancer Care. 2017;26: e12689. doi:10.1111/ecc.12689.

Q16: One of your key findings is around the stark difference between the patient and health professional perspectives. This deserves more prominence in the Discussion, since your main audience are health professionals who may find this challenging. You could refer more widely to other studies in different contexts – e.g. Young, J., Poole, U., Mohamed, F., Jian, S., Williamson, M., Ross, J., Jaye, C., Radue, P., & Egan, T. (2019). Exploring the value of social network 'care maps' in the provision of long-term conditions care. Chronic Illness. Advance online publication. doi: 10.1177/1742395319836463

R: We have emphasized the stark difference between the patient and healthcare professional perspective in the manuscript. We thank the reviewer for her suggested reference and have included it in the manuscript.

C: For healthcare professionals, it can be challenging to deliver person-centered care. MSA can serve as a tool to increase their understanding of people’s complex healthcare needs and identify duplications and gaps in their healthcare provision. The MSA also helps to remind them why they are in the caring profession and how they can provide patients with what they want and need. Previous research has shown that tools, like care mapping, have the potential to support the provision of person-centered care [46-48]. The MSA encourages reflection on the working methods of healthcare professionals and draws attention to the social situation of a patient, enabling healthcare professionals to provide person-centered care. (Discussion, page 21-22)

Additional references:

46. Chenoweth L, King MT, Jeon YH, Brodaty H, Stein-Parbury J, Norman R, et al. Caring for Aged Dementia Care Resident Study (CADRES) of person-centred care, dementia-care mapping, and usual care in dementia: A cluster-randomised trial. The Lancet Neurology. 2009;8(4): 317-325. doi:10.1016/S1474-4422(09)70045-6.

47. Crotty MM, Henderson J, Ward PR, Fuller J, Rogers A, Kralik D, et al. Analysis of social networks supporting the self-management of type 2 diabetes for people with mental illness. BMC Health Services Research. 2015;15:257. doi:10.1186/s12913-015-0897-x.

48. Young J, Poole U, Mohamed F, Jian S, Williamson M, Ross J, et al. Exploring the value of social network 'care maps' in the provision of long-term conditions care [published online ahead of print, 2019 Mar 18]. Chronic Illness. 2019:1-16. doi:10.1177/1742395319836463.

Abstract

Q17: Generally clear and an accurate representation of the paper, but add in a few more words about what MSA is, in line with the request about this in the Background section above.

R: We have added a few more words about MSA, in line with the additional information in the Background section.

C: The modular service architecture provides a comprehensive representation of the components and modules of healthcare provision. (Abstract, page 3)

Style/writing/references

Q18: In the main, the paper is structurally sound and reads well. Check again throughout for word choice and grammatically correct English. I have indicated below some of the more obvious corrections needed:

Line 51 – ‘Service modularity involves the decomposition of a complex service into modules…’ – the word ‘decomposition’ is probably better replaced with ‘deconstruction’, but check meaning and English language for clarity

Line 91 – ‘These four Downteams are well known in the field and provide a good of all Downteams in the Netherlands,…’ - correct the sentence

Line 109 – ‘The data were collected by VP…’ – explain who VP is – e.g. ‘author VP’ or ‘researcher VP’

Line 141 – ‘Three researched discussed and assessed the…’ – correct the sentence

Line 272-3 – ‘The modular perspective enabled us to fully decompose the healthcare provision into modules…’ – correct the word ‘decompose’ (similar problem to Line 51 comment above)

R: We have checked the manuscript for word choice and grammatically correct English. Decomposition is the right word to use here, since it is often used in the literature on modular service architecture. We present the obvious corrections as suggested by the reviewer in this response letter, but have made changes throughout the manuscript as a whole. 

C: These four Downteams are well-known in the field and demonstrate variety in their composition, working methods and geographic location, resulting in a comprehensive view on chronic healthcare for children with DS. (Methods, page 9)

The data were collected by researcher VP… (Methods, page 10)

The three researchers (VP, BM and EV) discussed and assessed the… (Methods, page 13)

Reviewer: 2 

Q19: The paper started to address an interesting and timely research area, focusing on providing person-centered care for patients with complex healthcare needs. The authors draw on some prior studies, but do not offer a very critical literature analysis. This is needed to strengthen the paper’s argument and draw out the gaps they seek to address. Also, the paper needs to present a much deeper methods and findings sections as well as stronger discussion and conclusion sections in order to offer value to the reader. Overall, the manuscript makes some very interesting points and I realise that a lot of work went into this study. Nevertheless, I see room for improvement which will help to enhance clarity, readability, practical and theoretical contributions. The following paragraphs address each section of the paper in more detail and provide suggestions on how to revise the paper.

R: We thank the reviewer for the compliment. We will go over the comments to clarify the issues point by point below.

Introduction

Q20: While the author(s) establish some links to some extant literature, author(s) need to establish a more coherent framework for the overall paper. That means, the introduction should clearly indicate the need for this paper in relation to extant research studies. Which gap(s) are authors trying to address here? In order to link back to relevant extant literature, work on services, service complexity (Kreye et al. and work by Andy Neely et al.) and customization/modularization needs to be further emphasized (which gaps you seek to address and what you found). This would help to further ground your study in extant work and help you to motivate your study further.

R: We understand the concerns of the reviewer and have been more thorough in explaining which gaps we seek to address. We have added a section to our introduction section in which we link back to extant research studies on services and service complexity. We address the gaps in Q21. We are grateful that the reviewer has suggested some key references and have added these, among other references, to further ground our study in extant work. Unfortunately, the work by Andy Neely et al. (2011) was inaccessible to us.

C: The complexity of healthcare services, an example of knowledge intensive professional services, stems from multiplicity and diversity in their service offering [9-11]. Multiplicity refers to the growing number of involved providers, components and interactions in service provision [11,12] and is demonstrated by the various professionals, from different units or departments, who deliver a high number of components for the treatment of patients with complex healthcare needs. This highly professionalized workforce needs to collaborate, something that could contradict the professional autonomy of the professionals [9,13] and, consequently, increase the complexity of healthcare services. Diversity refers to the growing variety of providers, components and interactions that are required to fulfill diversified patient needs [11]. Each patient has an individual constellation and combination of health problems which implies that multiple professionals are required to address these health problems. In addition, the steep information asymmetry between professionals and patients, a characteristic that is inherent to knowledge intensive professional services [9], can result in ambiguously expressed healthcare needs and increases the complexity of healthcare services. (Background, page 5)

Additional references:

9. von Nordenflyght A. What is a professional service firm? Toward a theory on taxonomy of knowledge-intensive firms. The Academy of Management Review. 2010;35(1): 155-174. doi:10.5465/amr.35.1.zok155.

10. Lewis MA, Brown AD. How different is professional service operations management? Journal of Operations Management. 2012;30: 1-11. doi:10.1016/j.jom.2011.04.002.

11. Zou W, Brax SA, Rajala R. Complexity in product-service systems: Review and framework. Procedia CIRP. 2018;73: 3-8. doi: 10.1016/j.procir.2018.03.319.

12. Kreye M, Roehrich JK, Lewis MA. Servitizing manufacturers: The importance of service complexity and contractual and relational capabilities. Production Planning & Control. 2015;26(14-15): 1233-1246. doi:10.1080/09537287.2015.1033489.

Conceptual background & Theoretical development

Q21: The authors need to establish some clearer links to extant literature (e.g. please see suggested key references). The author(s) should clearly draw out what is known for their key concepts. For instance, who is responsible for breaking up modules and components in your setting?; How easily can this be done?; How would this translate to other setting and what does prior work say about all this? This section needs much more information and guidance for reader to position the gaps in prior work. 

R: We have rewritten a major part of our introduction section in which we establish clearer links to the extant literature. We ensure that the reader gets a more detailed articulation of the gaps in prior work, emphasizing the importance of our study. 

C: Despite the potential of MSA to provide services that are responsive to the complex needs and requirements of customers, empirical evidence on the application of MSA is rare [19]. Although previous research provides examples of modular decomposition of healthcare services such as home care for the elderly [18], residential mental healthcare [20] and cancer care [21], these studies do not provide the complete MSA of these healthcare services. This results in an incomplete representation of the service offering and limits the potential of MSA to mix-and-match components and create truly individualized modular packages for each patient. Since only a few studies have addressed the applicability of MSA in complex services [13,22,23], there is still ambiguity around how to decompose a service offering into components and how to determine which of these components, alone or together, can be assigned as modules [24-26]. Dörbecker & Böhmann [27] have developed guiding questions that can guide the identification of components and modules for the creation of MSA, but these are only applied to a limited extent [3]. In addition, the few studies that do address the applicability of MSA are traditionally conducted from the professional’s perspective [13,22,23], which is surprising given the indispensable involvement of the customer in service provision [10,12,28].

In healthcare, the professional’s perspective mainly reflects the provision of healthcare services aimed at improving medical outcomes [1,13,23] and does not respond to the individual situations of people with complex healthcare needs. As a result, care is often not optimally tailored to their needs. The medical outcomes are often not the most relevant from a patient’s perspective; patients often attach greater value to functional outcomes and overall wellbeing [1,2]. However, it is increasingly acknowledged that insight into the patient’s perspective is becoming more important, especially for tailoring care to the needs and preferences of patients [1,2,29], quality of care [30, 31], and coordination of care [32,33]. These are all considered essential elements of person-centered care [34]. Our aim is therefore twofold. First, we provide the complete modular service architecture of healthcare provision for people with complex healthcare. This allows for the creation of individualized modular healthcare packages and supports the provision of person-centered care. Second, we provide insight into the patient’s perspective on MSA and explore how their perspective can support the provision of person-centered care. By doing so, we respond to the call for further empirical study on the application of MSA [19] and the call for more insight into the patients’ perspective on complex (modular) healthcare services [19,26,34].

We address these gaps in a multiple case study where we explore the applicability of MSA in healthcare provision for people with complex healthcare needs from the perspective of patients besides that of the healthcare professionals. We used chronic healthcare for children with DS as an example, and focused on the question whether MSA can support the provision of person-centered care. (Background, page 6-8)

Additional references:

18. de Blok C, Luijkx K, Meijboom B, Schols J. Improving long-term care provision: Towards demand-based care by means of modularity. BMC Health Services Research. 2010;10: 278-290. doi:10.1186/1472-6963-10-278.

20. Soffers R, Meijboom B, van Zaanen J, van der Feltz-Cornelis C. Modular health services: a single case study approach to the applicability of modularity to residential mental healthcare. BMC Health Services Research. 2014;14: 210-220. doi:10.1186/1472-6963-14-210.

21. Gobbi C, Hsuan J. Modularity in cancer care provision. In: van Donk PD, de Koster R, de Leeuw S, Fransoo J, van der Veen J, editors. EurOMA 2012: Proceedings of the 19th European Operations Management Association (EurOMA); 2012 Jul 1-5; Amsterdam, the Netherlands. Brussels: European Management Association.

22. Bask A, Merisala-Rantanen H, Tuunanen T. Developing a modular service architecture for e-store supply chains: The small- and medium-sized enterprise perspective. Service Science. 2014;6(4): 251-273. doi:10.1287/serv.2014.0082.

24. Salvador F, Forza C, Rungtusanatham M. Modularity, product variety, production volume, and component sourcing: Theorizing beyond generic prescriptions. Journal of Operations Management. 2002;20(5): 549-575 doi:10.1016/S0272-6963(02)00027-X.

25. Eissens–van der Laan M, Broekhuis M, van Offenbeek MAG, Ahaus CTB. Service decomposition: A conceptual analysis of modularizing services. International Journal of Operations & Production Management. 2016;36(3): 308-331. doi:10.1108/IJOPM-06-2015-0370.

26. Bartels EA, Meijboom BR, Nahar L, de Vries E. How service modularity can contribute to person-centered healthcare: a literature review. Paper presented at: EurOMA Doctoral Seminar, 27th Conference of the European Operations Management Association (EurOMA); 2020 Jun 25-26; Warwick, United Kingdom.

27. Dörbecker R, Böhmann T. Tackling the granularity problem in service modularization. In: Proceedings of the Twenty-first Americas Conference on Information Systems; 2015 Aug 13-15; Fajardo, Puerto Rico. New York: Curran Associates; 2016. p. 974-985. 

28. Cook LS, Bowen DE, Chase RB, Dasu S, Stewart DM, Tansik DA. Human issues in service design. Journal of Operations Management. 2002;20(2): 159-174. doi:10.1016/S0272-6963(01)00094-8.

29. Phelps RA, Pinter JD, Lollar DJ, Medlen JG, Bethell CD. Health care needs of children with Down syndrome and impact of health system performance on children and their families. Journal of Developmental & Behavioral Pediatrics. 2012;33(3):214-220. doi:10.1097/DBP.0b013e3182452dd8.

30. Minnes P, Steiner K. Parent views on enhancing the quality of health care for their children with fragile X syndrome, autism or Down syndrome: Parents’ perspectives. Child: Care, Health and Development. 2009;35(2): 250-256. doi:10.1111/j.1365-2214.2008.00931.x.

32. Singer SJ, Burgers J, Friedberg M, Rosenthal MB, Leape L, Schneider E. Defining and measuring integrated patient care: Promoting the next frontier in health care delivery. Medical Care Research and Reviews. 2011;68(1): 112-127. doi:10.1177/1077558710371485.

33. Miller AR, Condin CJ, McKellin WH, Shaw N, Klassen AF, Sheps S. Continuity of care for children with complex chronic health conditions: parents' perspectives. BMC Health Services Research. 2009;9:242. doi:10.1186/1472-6963-9-242.

34. Häkansson Eklund J, Holmström IK, Kumlin T, Kaminsky E, Skoglund K, Höglander J, Sundler AJ, Condén E, Summer Meranius M. “Same same or different?” A review of reviews of person-centered and patient-centered care. Patient Education and Counseling. 2019;102(1): 3-11. doi:10.1016/j.pec.2018.08.029.

Q22: Also, given the nature of the services you are investigating, the authors should at least acknowledge prior work on knowledge intensive professional (KIP) services and their unique characteristics and features.

R: We now acknowledge prior work on knowledge intensive professional (KIP) services and their unique characteristics and features. Please also see our reply to question 20.

C: This highly professionalized workforce needs to collaborate, something that could contradict the professional autonomy of the professionals [9,13] and, consequently, increase the complexity of healthcare services. (Background, page 5)

In addition, the steep information asymmetry between professionals and patients, a characteristic that is inherent to knowledge intensive professional services [9], can result in ambiguously expressed healthcare needs and increases the complexity of healthcare services. (Background, page 5)

As such, the MSA visualization provides a means of dealing with the complexity (i.e. multiplicity and diversity) of knowledge intensive professional services [9,11]. (Discussion, page 19)

Additional reference:

9. von Nordenflyght A. What is a professional service firm? Toward a theory on taxonomy of knowledge-intensive firms. The Academy of Management Review. 2010;35(1): 155-174. doi:10.5465/amr.35.1.zok155.

11. Zou W, Brax SA, Rajala R. Complexity in product-service systems: Review and framework. Procedia CIRP. 2018;73: 3-8. doi:10.1016/j.procir.2018.03.319.

Methods and Analysis

Q23: Overall, this section started to offer some insights, but I see room for improvements to obtain robust results and strengthen the paper’s argument. The authors should address the following concerns: 

Please clearly tell the reader about your sampling logic with regards to interviewees, context/sector and type of services.

R: We added more detail about our sampling logic with regards to interviewees, context/sector and type of services.

C: We aimed to select a range of available Downteams in the Netherlands that vary in their composition and working methods in order to select a representative set of Downteams. We used purposive sampling logic and carefully selected four out of the 22 Downteams in the Netherlands [38] to include in our research. These four Downteams are well-known in the field and demonstrate variety in their composition, working methods and geographic location, resulting in a comprehensive view on chronic healthcare for children with DS. As such, they provided a good representation of all Downteams in the Netherlands. (Methods, page 8-9)

Recruitment of participants was carried out by the coordinators (the pediatricians) of the Downteams based on purposive sampling logic. In the summer of 2017, they invited all the healthcare professionals in their Downteam and potentially interested parents of children with DS using e-mail, face-to-face requests and telephone. (Methods, page 9)

Q24: Here you need to also bring out the differences between interviewees/observations etc. At the moment, most of the data converge and no real differences are noticeable.

R: We bring out the differences between interviewees and observations more clearly in our manuscript.

C: The different types of data were complementary to each other: interviews helped us to acquire information on the professional's and patient’s perspective on care provision, observations allowed us to get a better impression of the daily practice of care provision, and the documents collected gave valuable information with regard to the composition and working methods of the Downteams. (Methods, page 12)

Q25: Clearly and in more detail explain your data analysis steps (what have you analysed first and how)

R: We added more detail about our data analysis steps to the manuscript.

C: The final data consisted of transcripts of the interviews, field notes and theoretical memos from the observations and documentation. (Methods, page 12)

the documents collected gave valuable information with regard to the composition and working methods of the Downteams. (Methods, page 12)

The participants did not express themselves in modularity terms, but instead we used modularity as a perspective that guided interpretation of the data. By combining the information from the interviews, observations and documentation, we were able to describe and interpret the practices provided by healthcare professionals in modular terms. For example: we used the national guideline [40] to assign distinct parts of the consultation from each individual professional as modules, as per our definition of modules [15]. The transcripts, field notes and theoretical memos were then used to corroborate the parts assigned as modules. We used guiding questions (e.g. for what are the modules used?, who will use these modules?) [27] to validate our interpretation of modules. If this differed from our interpretation, we reconsidered how the modules had been assigned. As a result, we went back and forth with all the collected data. We returned to our participants to prevent potential errors of interpretation [43]. The participants recognized the modular perspective in their way of working. (Methods, page 12)

Based on the information from the interviews, observations, and documentation, we described in modular terms the practices undertaken by the healthcare professionals in the four Downteams, using our coding of the text fragments as a basis. (Results, page 13)

Additional reference:

27. Dörbecker R, Böhmann T. Tackling the granularity problem in service modularization. In: Proceedings of the Twenty-first Americas Conference on Information Systems; 2015 Aug 13-15; Fajardo, Puerto Rico. New York: Curran Associates; 2016. p. 974-985.

43. Birt L, Scott S, Cavers D, Campbell C, Walter F. Member checking: A tool to enhance trustworthiness or merely a nod to validation? Qualitative Health Research. 2016;26(13): 1802–1811. doi:10.1177/1049732316654870.

Q26: A lot of the results section “tells” reader about your data rather than “shows” your data. Here, some data tables/figures and much more of a narrative would help to convince the reader that your findings support your conclusions.

R: We have rewritten some parts of the results section to convince the reader that our findings support our conclusions. We do not feel that data tables/figures would add to the comprehensibility of the paper because the MSA’s are the main focus of our results section. We belief that adding other tables/figures would distract the reader from the main focus. For clarity sake, we have not added the complete results section to the response letter. Please see our changes in results section of the manuscript accordingly.

Q27: What is the impact of different skills/capabilities of professionals when talking about/judging modules and components?

R: The professionals would not express themselves using modularity terms. However, when we returned to the professionals they argued that they recognized the modular perspective in their way of working. We added a sentence emphasizing this. 

C: Although the healthcare professionals did not express themselves in modular terms, they could recognize their way of working when presented with the MSA visualization. (Results, page 15)

Q28: Parents may struggle to judge (as we are talking about KIP services) the value of medical services. How can you really compare data from healthcare professionals and non-professionals? Wouldn’t you expect differences in perception and value?

R: We agree that there is a certain imbalance between what parents think they know and what they actually know. We added additional sentences emphasizing this.

C: It offers patients and professionals the possibility of mixing and matching person-centered modules and components to create individualized person-centered care packages without ignoring the professional role of the healthcare professionals. The extent to which each patient can create their own modular package is debatable: while some patients are clearly capable of this, it may be more difficult for others [9,13]. It would be difficult for patients with limited advocacy skills to create and arrange their own healthcare services [31]. (Discussion, page 21)

Q29: How have you measured the value (e.g. information exchange) of new/revised MSAs?

R: Measuring the value of the new/revised MSA fell beside the scope of this research. However, we agree with the reviewer that this is an important issue. We now suggest this as a direction for future research.

C: Lastly, we did not measure the value of the MSA approach. Future research could examine whether Downteams with a person-centered MSA approach are associated with better outcomes on process-indicators (e.g. adherence to guidelines, access to care) and outcome indicators (e.g. safety of care, patient satisfaction) compared with Downteams that use a traditional approach. (Discussion, page 24) 

Discussions and Conclusions

Q30: Derived from a conceptual background section which did not clearly draw out the gaps the paper seeks to address, the discussion and conclusion sections do offer only some additional value to the reader as it stands. The authors need to offer more fine-grained results here and discuss what they intended to find out in the introduction section (link to research question; overall aim of the paper). The authors should also link their findings to prior studies. Overall, the authors need to draw out clearly what the theoretical contributions are and how they add to the existing body of knowledge. This section also needs to clear link back to extant studies to offer some clear value to the reader. This should then help to derive some more insightful practice and policy implications.

R: We understand the concerns of the reviewer and we draw out more clearly what the theoretical contributions are and how they add to the existing body of knowledge. This helped us to derive more insightful practice and policy implications.

C: Previous studies only provided partial modular decompositions of healthcare services [18,20,21] which limited the potential of modularity to reduce service complexity and increase responsiveness to complex healthcare needs [14]. Our results show that the MSA proved to be very illuminating for professionals and patients since it led to insight into the work practices of each professional, which increased transparency on services offered for both professionals and patients. In addition, the MSA revealed gaps and overlaps in healthcare provision, and provided opportunities to deal with unnecessary duplications and blind spots. Moreover, we show that MSA visualization provides possibilities for mixing and matching components and modules to address individual needs and, as such, increases the responsiveness of healthcare services to people with complex healthcare needs. This demonstrates that MSA supports service customization [10,16,18], which can consequently lead to truly person-centered care provision [26,34]. As such, the MSA visualization provides a means of dealing with the complexity (i.e. multiplicity and diversity) of knowledge intensive professional services [9,11]. As distinct from previous studies on MSA [13,22,23], we have incorporated the patients’ perspective on MSA into our study and show that their perspective is essential for fulfilling the needs and preferences that are considered relevant by patients. The indispensable role of customers has been acknowledged in the extant service literature [e.g. 10,12,28], but the literature on service modularity has mostly overlooked this [19]. Our study shows that the patient’s perspective is essential to mix-and-match components in such a way that modular healthcare packages are created that are truly responsive to the needs and requirements of people with complex healthcare needs. As such, the service offering can be customized effectively [19] and the provision of person-centered care is supported [26,34]. (Discussion, page 19-20)

This person-centered MSA visualization provides a complete representation of the healthcare service based on functional outcomes and overall wellbeing and shows that insight into the patients’ perspective is important for the delivery of person-centered care [30,31,33]. While previous studies on modular decompositions implicitly assume that they fulfill patients’ needs and preferences [18,20], we show that the person-centered MSA can be used as a tool to ensure the complex healthcare needs of people with chronic conditions are fulfilled. (Discussion section, page 21)

Our proposed person-centered approach provides more clarity on how to identify the individual parts of a service offering and which part(s) can be considered as components or as modules, which is crucial for the modular decomposition of services [24-27]. By applying our approach, future studies can demonstrate the modular composition of their case under study and the insights obtained can become more relevant for theory as well as for practice. (Discussion, page 22)

Additional references:

9. von Nordenflyght A. What is a professional service firm? Toward a theory on taxonomy of knowledge-intensive firms. The Academy of Management Review. 2010;35(1): 155-174. doi:10.5465/amr.35.1.zok155.

10. Lewis MA, Brown AD. How different is professional service operations management? Journal of Operations Management. 2012;30: 1-11. doi:10.1016/j.jom.2011.04.002.

11. Zou W, Brax SA, Rajala R. Complexity in product-service systems: Review and framework. Procedia CIRP. 2018;73: 3-8. doi:10.1016/j.procir.2018.03.319.

12. Kreye M, Roehrich JK, Lewis MA. Servitizing manufacturers: The importance of service complexity and contractual and relational capabilities. Production Planning & Control. 2015;26(14-15): 1233-1246. doi:10.1080/09537287.2015.1033489.

18. de Blok C, Luijkx K, Meijboom B, Schols J. Improving long-term care provision: Towards demand-based care by means of modularity. BMC Health Services Research. 2010;10: 278-290. doi:10.1186/1472-6963-10-278.

20. Soffers R, Meijboom B, van Zaanen J, van der Feltz-Cornelis C. Modular health services: a single case study approach to the applicability of modularity to residential mental healthcare. BMC Health Services Research. 2014;14: 210-220. doi:10.1186/1472-6963-14-210.

21. Gobbi C, Hsuan J. Modularity in cancer care provision. In: van Donk PD, de Koster R, de Leeuw S, Fransoo J, van der Veen J, editors. EurOMA 2012: Proceedings of the 19th European Operations Management Association (EurOMA); 2012 Jul 1-5; Amsterdam, the Netherlands. Brussels: European Management Association.

22. Bask A, Merisala-Rantanen H, Tuunanen T. Developing a modular service architecture for e-store supply chains: The small- and medium-sized enterprise perspective. Service Science. 2014;6(4): 251-273. doi:10.1287/serv.2014.0082.

25. Eissens–van der Laan M, Broekhuis M, van Offenbeek MAG, Ahaus CTB. Service decomposition: A conceptual analysis of modularizing services. International Journal of Operations & Production Management. 2016;36(3): 308-331. doi:10.1108/IJOPM-06-2015-0370.

27. Dörbecker R, Böhmann T. Tackling the granularity problem in service modularization. In: Proceedings of the Twenty-first Americas Conference on Information Systems; 2015 Aug 13-15; Fajardo, Puerto Rico. New York: Curran Associates; 2016. p. 974-985.

28. Cook LS, Bowen DE, Chase RB, Dasu S, Stewart DM, Tansik DA. Human issues in service design. Journal of Operations Management. 2002;20(2): 159-174. doi:10.1016/S0272-6963(01)00094-8.

30. Minnes P, Steiner K. Parent views on enhancing the quality of health care for their children with fragile X syndrome, autism or Down syndrome: Parents’ perspectives. Child: Care, Health and Development. 2009;35(2): 250-256. doi:10.1111/j.1365-2214.2008.00931.x.

33. Miller AR, Condin CJ, McKellin WH, Shaw N, Klassen AF, Sheps S. Continuity of care for children with complex chronic health conditions: parents' perspectives. BMC Health Services Research. 2009;9:242. doi:10.1186/1472-6963-9-242.

34. Häkansson Eklund J, Holmström IK, Kumlin T, Kaminsky E, Skoglund K, Höglander J, Sundler AJ, Condén E, Summer Meranius M. “Same same or different?” A review of reviews of person-centered and patient-centered care. Patient Education and Counseling. 2019;102(1): 3-11. doi:10.1016/j.pec.2018.08.029.

---

## [Decision Letter · Decision Letter 1]

24 Sep 2020

PONE-D-20-03347R1

Providing person-centered care for patients with complex healthcare needs: A qualitative study

PLOS ONE

Dear Dr. Meijboom,

Thank you for submitting your manuscript to PLOS ONE. After careful consideration, we feel that it has merit but does not fully meet PLOS ONE’s publication criteria as it currently stands. Therefore, we invite you to submit a revised version of the manuscript that addresses the points raised during the review process.

We look forward to receiving your revised manuscript.

Kind regards,

Janhavi Ajit Vaingankar

Academic Editor

PLOS ONE

Reviewers' comments:

Reviewer's Responses to Questions

**Comments to the Author**

1. If the authors have adequately addressed your comments raised in a previous round of review and you feel that this manuscript is now acceptable for publication, you may indicate that here to bypass the “Comments to the Author” section, enter your conflict of interest statement in the “Confidential to Editor” section, and submit your "Accept" recommendation.

Reviewer #2: All comments have been addressed

2. Is the manuscript technically sound, and do the data support the conclusions?

Reviewer #2: Yes

3. Has the statistical analysis been performed appropriately and rigorously? 

Reviewer #2: N/A

4. Have the authors made all data underlying the findings in their manuscript fully available?

Reviewer #2: Yes

5. Is the manuscript presented in an intelligible fashion and written in standard English?

Reviewer #2: Yes

6. Review Comments to the Author

Reviewer #2: Many thanks for addressing all my comments from the previous round. I would suggest that the authors address the following comments in their next round of revisions:

# please proof-read the whole manuscript to address typos and spelling mistakes

# please ensure that all references are complete and formatted consistently

# Please discuss in more detail how your findings may be applied to other types of (healthcare) services to offer some degree of generalisability and links to possible future research

7. PLOS authors have the option to publish the peer review history of their article (what does this mean?). If published, this will include your full peer review and any attached files.

Reviewer #2: No

---

## [Author Response · Author response to Decision Letter 1]

27 Oct 2020

Reviewer: 2 

Q1: Many thanks for addressing all my comments from the previous round. I would suggest that the authors address the following comments in their next round of revisions.

R: We are happy to hear that we addressed the comments in an appropriate manner. We will go over the remaining comments to clarify the issues point by point below.

Q2: Please proof-read the whole manuscript to address typos and spelling mistakes

R: We have addressed an experienced copy editor to proofread the whole manuscript. The copy editor is a native English speaker and she has addressed the typos and spelling mistakes. Please see the manuscript accordingly.

Q3: Please ensure that all references are complete and formatted consistently

R: We have ensured that all references are complete and formatted consistently. Please see the reference list for our changes.

Q4: Please discuss in more detail how your findings may be applied to other types of (healthcare) services to offer some degree of generalisability and links to possible future research

R: We incorporated more detail about how our findings can be applied in other types of (healthcare services) to offer some degree of generalizability and links to possible future research. We now provide two examples of other types of services: legal services and higher educational services.

C: We also believe that the MSA approach is applicable in many other types of complex services such as legal services or higher educational services. For example, when clients face a legal conflict (divorce, termination of employment etc.) they can make use of a variety of providers in dealing with their conflict. Each provider is responsible for providing a subset of services for the client and collectively the providers offer the service that fits with the client’s needs and wishes. Approaching legal services from a modular perspective allows for the decomposition into components and modules, resulting in transparency on the supply side of legal services. The modular perspective is also relevant for clients because although accurate information about legal services becomes increasingly available online [52], the clients are not fully aware of what each legal provider can offer them. This results in legal services that are not completely tuned to the needs and requirements of clients. In higher educational services in many countries there has been an increasing focus on individualized instructions, despite the increasing number of students [53]. As a result, there is a need to make higher education services available to large number of students and, at the same time, offer an individualized learning package for each student. A modular approach could help in dealing with this issue, as it provides opportunities to offer curricula or interdisciplinary programs that are designed based on modules, where each student’s program is tailored to their individual needs and wishes [54]. Future research should test whether the MSA approach is feasible in these complex service settings. (Discussion, page 23-24)

Additional reference:

52. Giannakis M, Doran D, Mee D, Papadopoulos T, Dubey, R. The design and delivery of modular legal services: Implications for supply chain strategy. International Journal of Production Research. 2018;56(20): 6607-6627. doi:10.1080/00207543.2018.1449976

53. Goldschmid B, Goldschmid ML. Modular instruction in higher education: A review. Higher Education. 1973;2: 15-32. doi:10.1007/BF00162534

54. Turnbull W, Burton D, Mullins P. Strategic repositioning of institutional frameworks’: Balancing competing demands within the modular UK higher education environment. Quality in Higher Education. 2008;14(1): 15-28. doi:10.1080/13538320802011474

---

## [Editor Report · Decision Letter 2]

3 Nov 2020

Providing person-centered care for patients with complex healthcare needs: A qualitative study

PONE-D-20-03347R2

Dear Dr. Meijboom,

We’re pleased to inform you that your manuscript has been judged scientifically suitable for publication and will be formally accepted for publication once it meets all outstanding technical requirements.

Kind regards,

Janhavi Ajit Vaingankar

Academic Editor

PLOS ONE

---

## [Editor Report · Acceptance letter]

6 Nov 2020

PONE-D-20-03347R2 

Providing person-centered care for patients with complex healthcare needs: A qualitative study 

Dear Dr. Meijboom:

I'm pleased to inform you that your manuscript has been deemed suitable for publication in PLOS ONE. Congratulations! Your manuscript is now with our production department. 

Kind regards, 

on behalf of

Ms Janhavi Ajit Vaingankar 

Academic Editor

PLOS ONE